# A Retrospective Study of the Impact of COVID-19 Pandemic Related Administrative Restrictions on Spine Surgery Practice and Outcomes in an Urban Healthcare System

**DOI:** 10.3390/ijerph191710573

**Published:** 2022-08-25

**Authors:** Bahar Attaripour, Selena Xiang, Brendon Mitchell, Matthew Siow, Jesal Parekh, Bahar Shahidi

**Affiliations:** Department of Orthopaedic Surgery, University of California San Diego, 9500 Gilman Drive MC0863, La Jolla, CA 92037, USA

**Keywords:** spine, COVID-19 pandemic, surgery, orthopaedic surgery

## Abstract

The study objective is to characterize the impact of COVID-19 related hospital administrative restrictions on patient demographics, surgical care, logistics, and patient outcomes in spine surgery. This was a retrospective study of 331 spine surgery patients at UCSD conducted during 1 March 2019–31 May 2019 (pre-COVID-19) and 1 March 2020–31 May 2020 (first COVID-19 surge). All variables were collected through RedCap and compared between pre- and during-COVID groups. There were no significant differences in patient demographics, operating room duration, and skin-to-skin time. However, length of stay was 4.7 days shorter during COVID-19 (*p* = 0.03) and more cases were classified as ‘urgent’ (*p* = 0.04). Preoperative pain scores did not differ between groups (*p* = 0.51). However, pain levels at discharge were significantly higher during COVID (*p* = 0.04) and trended towards remaining higher in the short- (*p* = 0.05) but not long-term (*p* = 0.17) after surgery. There was no significant difference in the number of post-operative complications, but there was an increase in the use of the emergency room and telemedicine to address complications when they arose. Overall, the pandemic resulted in a greater proportion of ‘urgent’ spine surgery cases and shorter length of hospital stay. Pain levels upon discharge and at short-term timepoints were higher following surgery but did not persist in the long term.

## 1. Introduction

The COVID-19 pandemic has substantially changed how healthcare workers provided medical care. Due to the changing dynamics for hospitals, healthcare-workers, and patients alike, the prioritization of surgical cases, the risk status of the patient, and the availability of resources to administer high quality health care has been modified.

The pandemic’s early effect on orthopaedic surgery was particularly unique as most cases are modifiable, due to the elective nature of many surgeries for musculoskeletal conditions. During the first wave of pandemic, physicians were often required to follow modified surgery decision algorithms, resulting in the postponement of elective surgeries to a later time [1,2]. Along with surgery postponement, there is some evidence that patients were hesitant to come to the hospital due to the perceived risk of contracting the COVID-19 virus in overburdened health care settings [3,4]. Deferring non-urgent cases, due to administrative restrictions in the form of surgical rationing or patient hesitancy may influence the overall health risk of patients the longer a procedure is postponed [5,6,7,8]. Specifically, reductions in quality of life during the postponement period, as well as the impact of postponement on disease progression, possibly lead to the need for a more complex and expensive surgery with longer recovery time and poorer outcomes [5,9,10].

Most current literature describes administrative changes in surgical decision-making guidelines used to identify essential surgeries [11,12,13,14,15,16,17,18,19,20,21,22]. However, the impact of these guidelines on short and long-term patient outcomes in individuals undergoing surgery for musculoskeletal conditions of the spine has not been well described. Studies comparing case frequency and length of stay in individuals undergoing surgery before and during early phases of COVID-19 have shown that overall number of surgeries were reduced, with no differences in length of stay [11,23,24]. However, surgical outcomes were not assessed.

Therefore, the purpose of this study is to compare demographic, surgical, procedural, and outcomes-based characteristics of patients undergoing surgical care of the spine prior to the COVID-19 pandemic and during the first surge of the COVID-19 pandemic. We expected to see similar demographic outcomes, an increase in urgent and trauma cases due to the administrative restrictions on performing elective surgery during the first phase of the pandemic, as well as an increase in short- and long-term pain scores.

## 2. Materials and Methods

### 2.1. Demographic Data

This was a retrospective study of electronic medical information from patients who had spine surgery performed through UC San Diego Orthopaedic Surgery department from 1 March 2019 to 31 May 2019 (pre-COVID-19) and 1 March 2020 to 31 May 2020 (during the first COVID-19 surge). The first surge of the COVID-19 pandemic was chosen because this was the phase in which the most restrictive administrative restrictions on surgical practice occurred at our institution, resulting in severe surgical rationing and cessation of elective spinal surgeries. This study was reviewed and approved by the UCSD Institutional Review Board (#191676) and is in accordance with the Declaration of Helsinki. Operative logs within the periods of interest were used to generate a list of eligible cases for review. The electronic medical records (EMR) were queried to extract demographic data (age, gender, race, ethnicity, body mass index (BMI), American Society of Anesthesiologists (ASA) rating), risk factors (smoking, diabetes), payor information, surgical data (procedure name, discharge disposition, underlying diagnosis, length of stay, estimated blood loss), operational data (surgery time, recovery time, preparation time, follow up care type and quality), and outcomes (number and type of intraoperative and postoperative complications, patient-reported outcomes), which were captured in a RedCap database.

### 2.2. Surgical Data

Surgical data included procedure type, servicing department (trauma, spine, or other), procedure classification (inpatient vs outpatient), procedure urgency (urgent or elective), diagnostic indication, duration in operating room, skin-to-skin duration, estimated blood loss, hospital length of stay, and discharge disposition. Spine procedures were classified as one or more of the following: laminectomy, decompression, non-traumatic fixation/fusion, traumatic fixation/fusion, kyphoplasty, foraminectomy, discectomy, hardware removal, and other. The number of procedures within a given surgery and whether a surgery was staged (completed under two separate episodes of anesthesia) was also documented. In cases of multiple anesthesia events, the procedures were identified as staged and considered separately for analysis for all variables except length of stay, unless the patient left the hospital between stages. Diagnostic indications were broadly characterized based on ICD-9 or ICD-10 codes for the following conditions—spondylosis/stenosis, spondylolisthesis, scoliosis/kyphoscoliosis, tumor/infection, and fracture. Procedures were classified as inpatient if length of stay was greater than or equal to 24 h. Case urgency classifications were based on diagnosis and the presence/absence of red flag symptoms. Acute traumas resulting in unstable fractures, spinal cord compression, and/or 3 column injuries were deemed urgent. Patients presenting with signs/symptoms and imaging consistent with cauda equina syndrome were categorized as urgent. Spine infections were similarly classified as urgent, particularly in the post-operative setting and those in which there were concerns for evolving sepsis. Lastly, patients presenting with progressive neurologic deficits secondary to their spine pathology were deemed urgent. All other cases were categorized as elective, including degenerative pathologies, such as degenerative disc disease, coronal/sagittal imbalance, and spondylolisthesis.

### 2.3. Post-Operative Outcomes Data

Number and type of post-operative complications, 90-day readmission or emergency room visit, and reoperation were documented. Post-operative complications were defined as the following: (1) excessive pain, (2) infection, (3) hardware failure, (4) wound dehiscence, (5) reaction to anesthesia, (6) reaction to postoperative medications, or (7) other. Number and type (in-clinic or telemedicine) of postoperative follow up visits and number of in-patient physical therapy visits was also documented. Finally, patient reported outcomes for pain using the visual analogue scale were collected from follow up visits at the following timepoints: (1) pre-operative, (2) day of discharge, (3) short-term (<4 months) follow up, and (4) long-term (9–15 months) follow up.

### 2.4. Statistical Analysis

Patient demographics, surgical data, operational data, and outcomes measures were compared between pre- and during-COVID groups using independent t-tests for continuous variables, or chi-square tests for categorical or binary variables. Normality of data was assessed using Levene’s test.

## 3. Results

### 3.1. Demographic Results

Most patients undergoing spine surgery were in their sixth decade of life, were Caucasian, overweight (BMI > 25), diabetic, in ASA class 3, and never smokers (Table 1 and Table 2). There were no significant differences in patient demographics including age, BMI, gender, racial and ethnicity distribution, smoking or diabetes status, or ASA rating before and during the pandemic.

### 3.2. Surgical Results

There were 185 patients who underwent spine surgery in the pre-pandemic timeframe, whereas 146 patients underwent surgery during COVID-19 (a 21.1% reduction). Non-traumatic fixation/fusions were the most common procedures followed by laminectomy and discectomy prior to the COVID-19 pandemic (Figure 1). Significant increases in the proportion of patients undergoing laminectomy (14.7%, *p* = 0.006), unspecified decompression (22.9%, *p* < 0.001), kyphoplasty (7.7%, *p* = 0.030), foraminectomy (13.9%, *p* < 0.001), and/or discectomy (13.8%, *p* = 0.006) were observed during the COVID-19 pandemic, and there was a trend for an increase in proportion of traumatic fixation/fusions (5.1%, *p* = 0.098) (Figure 1). There were no significant changes in the proportion of non-traumatic fixation/fusions before and during COVID-19 (6.4%, *p* = 0.258). There was a larger proportion of patients undergoing four or more procedure types within a surgery (14% increase) and a smaller proportion of patients undergoing 2–3 procedure types within a surgery (14.7% decrease) during the pandemic (*p* < 0.001, Table 3). Despite the increase in the number of procedures performed within a given surgery during the pandemic, there was no difference in operating room duration (mean (SD) 317.9 (158.5) min) and skin-to-skin time (223.7 (137.2) min) between patients who had surgery prior versus during the pandemic (*p* > 0.641). However, we did observe a trend for an increase in estimated blood loss (mean difference 48 mL’s) during the pandemic (*p* = 0.064) (Table 3).

Spondylosis/stenosis was the most prevalent diagnostic indication for surgery (Figure 2). However, we observed a 5.4% (non-significant) reduction in patients with this indication during the pandemic (*p* = 0.062). We also observed a 10.6% reduction in patients with spondylolisthesis (*p* < 0.001), a 3.5% reduction in patients with scoliosis/kyphoscoliosis (*p* = 0.012), a 2.7% reduction in patients with tumor/infection (*p* < 0.001), and a 4.4% increase in patients with fracture (*p* < 0.001) during the pandemic as compared to prior (Figure 2). There was a trend for a reduction in the proportion of staged surgeries during the pandemic (7.3%, *p* = 0.072), and an increase in both the proportion and total number of cases designated as ‘urgent’, despite overall reductions in case numbers during the pandemic (9.9%, *p* = 0.035).

### 3.3. Operational and Patient Outcome Results

Length of stay was 4.66 days shorter during the pandemic as compared to prior to the pandemic (*p* = 0.032) along with a non-significant reduction in the number of in-patient physical therapy visits (*p* = 0.238). Most patients were discharged home both pre- and during-COVID-19, with no significant difference between groups (*p* = 0.400). There were no differences in postoperative complications within 90 days (*p* = 0.207). There was a statistically significant lower reoperation rate (*p* < 0.01) and 90-day readmission rate (*p* = 0.04) during COVID-19. However, the number of emergency room (ER) visits for complications increased (*p* = 0.048), as well as the number of telemedicine visits (*p* < 0.001). There were no differences in the number of in-clinic visits for spine-related complications within 90 days of surgery (*p* = 0.439) (Table 4). Preoperative pain scores did not differ between groups (*p* = 0.58). However, pain levels at discharge were significantly higher in patients undergoing surgery during COVID-19 (*p* = 0.04). Pain levels at discharge demonstrated a trend towards remaining higher in the short-term (*p* = 0.06), but not long-term (*p* = 0.21) after surgery (Figure 3). The majority of short-term visits for patients were at 6 weeks, 8 weeks, and 12 weeks, while long term follow-up visits ranged within 9–15 months (±2 weeks). However, approximately only 47% of patients in both the Pre-COVID-19 group and COVID-19 group obtained long term data for pain scores.

## 4. Discussion

The COVID-19 pandemic brought upon many changes to patient care. Our study demonstrated that there were no differences in the demographic characteristics of the patient population accessing care from before the pandemic to during the first wave of the pandemic. However, during this first wave, patients were more likely to be admitted for an urgent procedure (particularly for fractures) likely due to the changes in administrative requirements favoring urgent over elective surgeries, in addition to the shortage of hospital beds and risk of viral transmission. Moreover, patients underwent a greater number of coded procedures during each visit to the operating room, potentially accounting for the greater blood loss. Yet, despite the higher procedure density during each case, surgical time remained unchanged, and length of stay was shorter by approximately half (54%). Furthermore, complication rates remained the same, although patients reported greater pain on the day of discharge. Difference in pain scores did not persist in the long term (one year). The short-term increase in pain scores may be attributed to shorter hospital lengths of stay, since COVID-19 patients were closer in time to the surgery on the day of discharge, and possibly missed out on many in-patient post-operative pain management modalities normally administered over the duration of in-patient stay, thus potentially reporting higher levels of pain. This study is the first that investigates the long-term impact on patient outcomes as a result of administrative restrictions placed on urban hospital systems during the first surge of the COVID-19 pandemic.

Most prior investigations of the influence of the COVID-19 pandemic on spine surgery have focused on describing the administrative recommendations relative to patient selection for surgery. However, a small number of studies have compared demographic and some surgical indications data. Of those studies, most did not find differences in demographic characteristics between pre- and during-COVID-19 [24,25]. However, one study did show a significant change in gender and age, with those receiving spine surgery being significantly younger, and more male during the pandemic [23]. This was primarily attributed to a substantial reduction in percutaneous vertebroplasty procedures, which were more commonly performed in older females in the pre-COVID-19 comparison group [23]. Overall, the lack of change in demographic data suggests that accessibility to care, or racial and ethnic disparities in care hesitancy was not significantly impacted during the initial phases of the COVID-19 pandemic as a result of administrative restrictions on spine surgery in this patient population. Despite our observations, the possibility that these accessibility factors may play a greater role as the pandemic continued past the first wave may result in an underestimation of the overall influence of the pandemic on care accessibility and hesitancy in this study.

Related to surgical indications, our data confirms our hypothesis that administrative restrictions on performing elective surgery during the first wave of the pandemic resulted in a larger proportion of cases designated as urgent, and an increase in the proportion of individuals admitted for fracture (trauma). These observations are consistent with one study that observed an increase in urgent cases, although the sample size was less than 20 [12]. However, it contradicts another study that reported a decrease in spine fractures and an increase in infection and tumor indications during the pandemic [23]. This could be due to differences in the healthcare settings studied, as the study was performed in Italy and described patient populations specific to oncologic and degenerative spine surgery departments. This setting is likely to be more heavily geared toward treatment of tumor and degenerative conditions as opposed to traumatic injury compared to a United States-based hospital with a Level 1 trauma center.

One surprising finding was that although the number of surgical procedures for a given patient increased and the length of stay decreased substantially, there were no differences in discharge disposition or complication rates. This is consistent with prior literature, although those same studies did not observe the same reduction in length of stay and post-operative complications that we describe in the current study [23,24]. This suggests that the administrative efforts to optimize care efficiency were effective without negatively impacting patient risk in the immediate postoperative period, although the differences in patient selection and surgical indication may also influence these results. These administrative recommendations to reduce in-person encounters resulted in the reduction in in-clinic visits, and increased telemedicine visits when complications did arise. This increase in ER visits observed may have also been because patients were more inclined or encouraged to use the ER services to be seen in-person when routine in-person follow up care was less available. While we did observe that patients were discharged faster and with a higher pain level during the pandemic, these differences were less than 1 point in magnitude and thus likely not clinically significant. Importantly, these differences did not persist in the long term (9–15 months).

Our study was not without limitations. First, our study utilized a retrospective study design given that a prospective study design was not feasible due to the lack of prior knowledge of the course and nature of the pandemic. As such, many patients were lost to follow up and we were not able to retrieve complete data on them. Despite similar rates of loss to follow up in both groups, this may have resulted in underestimation of important complications for the pandemic group, and a reduction in data for one-year follow up visits for the pre-pandemic group due to their visits coinciding with the first wave of the pandemic. Second, although the most restrictive administrative changes at our institution occurred during the first wave of the COVID-19 pandemic, the subsequent waves may have had differential effects on surgical practices and patient outcomes. Future studies are required to continue to follow patients for longer periods and throughout the pandemic to evaluate whether longer term outcomes, such as reoperation rate, are influenced by the changes in care paradigms applied during the pandemic.

## 5. Conclusions

The pandemic resulted in significant changes in the surgical care of individuals with spinal pathology. Primarily, we observed that during the first wave of the pandemic there was a larger proportion of urgent surgeries, particularly fractures. We also observed that patients were discharged from the hospital 4.7 days sooner, with greater levels of pain. There were no differences in complication rates or long-term pain outcomes. However, patients demonstrated increased utilization of the ER and telemedicine to address complications when they did arise. More research is needed to determine whether these changes influence longer term patient outcomes.

## Figures and Tables

**Figure 1 ijerph-19-10573-f001:**
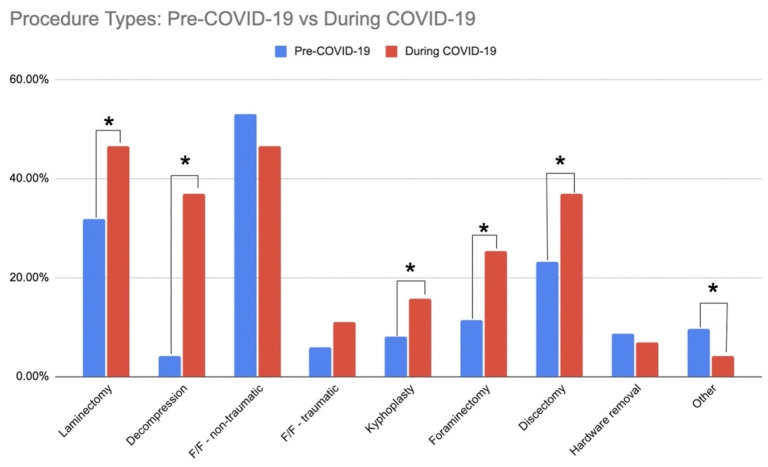
Procedure Types: Pre COVID-19 vs. During COVID-19. Significant group differences are indicated with an asterisk (*).

**Figure 2 ijerph-19-10573-f002:**
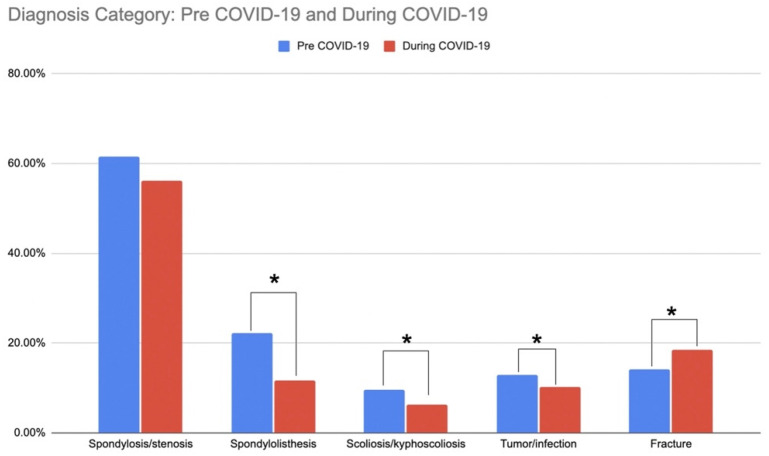
Diagnosis Category: Pre COVID-19 vs. During COVID-19. Significant group differences are indicated with asterisks (*).

**Figure 3 ijerph-19-10573-f003:**
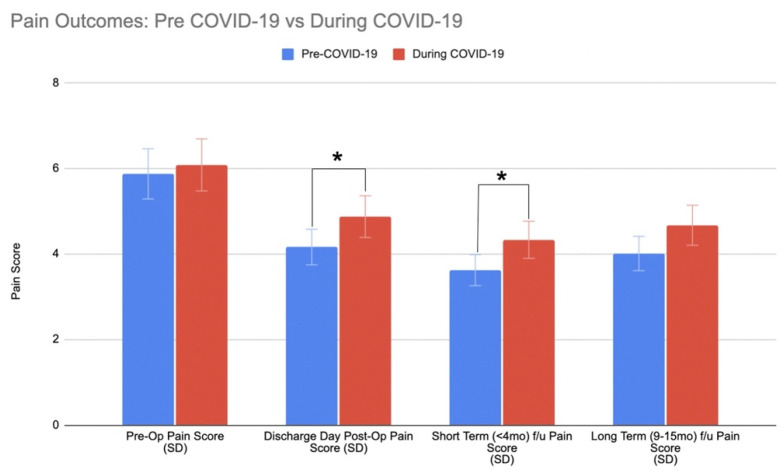
Pain Outcomes. Significant group differences are indicated with an asterisk (*).

**Table 1 ijerph-19-10573-t001:** Comparison of Demographic Data Pre COVID-19 and During COVID-19.

Demographics		Combined	Pre-COVID-19	During COVID-19	*p*-Value
**Age:** **Mean *(SD)***		60.6 (14.6)	60.97 (14.6)	60.16 (14.7)	0.617
**BMI** **Mean *(SD)***		28.3 (5.9)	28.7 (6.4)	27.7 (5.4)	0.144
**Gender** **N (%)**	Female	129 (39.0%)	76 (41.1%)	53 (36.3%)	0.439
	Male	201 (60.7%)	108 (58.4%)	93 (63.7%)	
	Other	0 (0.0%)	0 (0.0%)	0 (0.0%)	
	No Response	1 (0.3%)	1 (0.5%)	0 (0.0%)	
**Race** **N (%)**	White	204 (61.6%)	107 (57.8%)	97 (66.4%)	0.135
	Asian	19 (5.7%)	15 (8.1%)	4 (2.7%)	
	Black or African American	14 (4.2%)	9 (4.9%)	5 (3.4%)	
	Native Hawaiian or other Pacific Islander	2 (0.6%)	1 (0.5%)	1 (0.7%)	
	American Indian or Alaska Native	2 (0.6%)	0 (0.0%)	2 (1.4%)	
	Other Race or Mixed Race	88 (26.6%)	51 (27.6%)	37 (25.3%)	
	Unknown	2 (0.6%)	2 (1.1%)	0 (0.0%)	
**Ethnicity** **N (%)**	Non-Hispanic	262 (79.1%)	153 (82.7%)	109 (74.7%)	0.074
	Hispanic	69 (20.8%)	32 (17.3%)	37 (25.3%)	
**Insurance Type** **N (%)**	Medicare/Medicaid	207 (62.5%)	114 (61.6%)	93 (63.7%)	0.628
	Private	92 (27.8%)	51 (27.6%)	41 (28.1%)	
	Veterans	24 (7.3%)	15 (8.1%)	9 (6.2%)	
	Worker’s Comp	1 (0.3%)	0 (0.0%)	1 (0.7%)	

**Table 2 ijerph-19-10573-t002:** Comparison of Surgical Risks Pre COVID-19 and During COVID-19.

Risks		Combined	Pre-COVID-19	During COVID-19	*p*-Value
**ASA Rating** **N (%)**	1201	12 (3.6%)	4 (2.2%)	8 (5.5%)	0.143
	1202	131 (40.0%)	79 (42.7%)	52 (35.6%)	
	1203	165 (50.0%)	88 (47.6%)	77 (52.7%)	
	1204	17 (5.1%)	12 (6.5%)	5 (3.4%)	
**Smoking Status** **N (%)**	Never Smoker	172 (52.0%)	94 (50.8%)	78 (53.4%)	0.480
	Former Smoker	104 (31.4%)	59 (31.9%)	45 (30.8%)	
	Light Tobacco Smoker	3 (0.9%)	2 (1.1%)	1 (0.7%)	
	Current Some Day Smoker	15 (4.5%)	7 (3.8%)	8 (5.5%)	
	Current Everyday Smoker	24 (7.3%)	17 (9.2%)	7 (4.8%)	
	Never Assessed	11 (3.3%)	6 (3.2%)	5 (3.4%)	
	Unknown If I ever smoked	2 (0.6%)	0 (0.0%)	2 (1.4%)	
**Diabetes Status** **N (%)**	Yes	289 (87.3%)	163 (88.1%)	126 (86.3%)	0.624
	No	42 (12.7%)	22 (11.9%)	20 (13.7%)	

**Table 3 ijerph-19-10573-t003:** Comparison of Surgical Characteristics Pre COVID-19 and During COVID-19. Significant *p*-values are indicated in bold. # = number.

Surgical		Combined	Pre-COVID-19	During COVID-19	*p*-Value
**# of Procedure Types** **N (%)**	2 or 3	306 (91.6%)	183 (98.9%)	123 (84.2%)	**<0.001**
	4 or more	24 (8.1%)	2 (1.1%)	22 (15.1%)	
**Staged Surgery** **N (%)**	Yes	52 (15.3%)	35 (18.9%)	17 (11.6%)	0.072
	No	266 (80.8%)	143 (77.3%)	123 (84.2%)	
**Length of Stay** ** *(SD)* **		7.9 (17.6)	10.23 (28.3)	5.57 (6.8)	**0.032**
**Type of Service** **N (%)**	General Trauma	20 (6.0%)	13 (7.0%)	7 (4.8%)	0.378
	Spine	300 (90.6%)	167 (90.3%)	133 (91.1%)	
	Other	1 (0.2%)	0 (0.0%)	1 (0.7%)	
**Duration in OR** ** *(SD)* **		313.7 (167.3)	317.9 (158.5)	309.4 (176.1)	0.678
**Duration skin-to-skin** ** *(SD)* **		220 (140.9)	223.7 (137.2)	216.3 (144.6)	0.641
**Inpatient** **N (%)**	Yes	231 (70.0%)	163 (88.1%)	68 (46.6%)	**<0.001**
	No	100 (30.2%)	22 (11.9%)	78 (53.4%)	
**Type of Case** **N (%)**	Non-Urgent	254 (76.7%)	150 (81.1%)	104 (71.2%)	**0.035**
	Urgent	77 (23.3%)	35 (18.9%)	42 (28.8%)	
**Estimated Blood Loss *(SD)***		149.6 (219.4)	125.7 (174.6)	173.5 (264.2)	0.064
**Discharge Disposition** **N (%)**	Home	241 (73.1%)	131 (70.8%)	110 (75.3%)	0.400
	Nursing Facility	61 (18.5%)	34 (18.4%)	27 (18.5%)	
	Other	20 (5.9%)	14 (7.6%)	6 (4.1%)	

**Table 4 ijerph-19-10573-t004:** Comparison of Complications and Different Types of Visits Post Surgery between Pre- and During COVID-19. Significant *p*-values are indicated in bold.

Operations	Combined	Pre-COVID-19	During COVID-19	*p*-Value
**90 day post-operative complication rate**	76 (25.9%)	48 (28.7%)	28 (22.2%)	0.207
**Number of ER visits for spine-related complications w/in 90 days after surgery** ** *(SD)* **	0.62 (0.75)	0.49(0.70)	0.87(0.82)	**0.048**
**Number of Clinical visits for spine-related complications w/in 90 days after surgery** ** *(SD)* **	3.26 (1.60)	3.36 (1.73)	3.04 (1.33)	0.439
**Number of Telemedicine visits for spine-related complications w/in 90 days after surgery** ** *(SD)* **	0.10 (0.39)	0.00 (0.00)	0.32 (0.65)	**0.001**
**Number of inpatient PT interactions**	2.16 (3.7)	2.36 (4.50)	1.90 (2.26)	0.238

## Data Availability

Data will be made available upon reasonable request to the corresponding author.

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
