# Peer review of "A Retrospective Study of the Impact of COVID-19 Pandemic Related Administrative Restrictions on Spine Surgery Practice and Outcomes in an Urban Healthcare System"

_ijerph, 2022, doi:10.3390/ijerph191710573_

Round 1

Reviewer 1 Report

We congratulate the authors on a well conducted &  researched study .The authors have performed a retrospective review to analyze the impact of the first phase of covid 19 pandemic on spine surgery in a urban setup .   The authors have compared the  spine surgery practice to a year before during the same time . They noted more urgent admissions , shorter hospital stay & hence more pain level at the time of discharge .  Howevr the authors have noted no significant difference during follow up . What is surprising is that there was no difference in the operating time duration .   With lot of precautions being taken it is quite natural to  presume that the operating time duration would be more . 

The paper is an addition to the growing body of literature on th effect of Covid 19 on spine surgery .  The paper only looks at the first phase  of covid 19 pandemic . The Significant impact of covid 19 in USA would have been  after April 2022 & it would have been better to started the study from April 1 .   The significant impact of covid 19 is also  mainly during the second phase & hence the study should  have compared the effect during all the phases & more so during the second phase . 

The introduction . material methods & results , discussion is very well written .

Author Response

The manuscript is over-all well organized, well written with respect to language and structure. The study has been conducted on the effect of covid 19 on spine surgery practice . The impact of covid 19 as I understand would be after march 2020 &hence it would have be ideal to include the patients from that time period . Also the significant impact of covid 19 was during the 2nd  phase & that also should have been analysed .

Thank you for this comment. We agree that a larger analysis documenting the influence of the COVID-19 pandemic throughout the following phases would add value to this manuscript. There were two reasons that we did not include 2nd or subsequent wave data. First, we did not have complete follow up data for the retrospective review at the time of IRB approval for the 2nd wave. Second, and more importantly, our healthcare institution only implemented substantial administrative restrictions on elective surgeries during the first wave of the pandemic. At the time that the second wave began, our institution had ceased the surgical rationing procedures that we hypothesized would most impact the clinical outcomes. As such, we felt that a more accurate assessment of the impact of administrative changes was contained in the data limited to the first wave. We have now clarified this in our introduction and discussion sections.

Title -  Inappropriate . Does not include the important phases of covid 19 epidemic. As mentioned earlier the main effect of covid in USA was after march.

Thank you. As per the previous comment, we feel that a more accurate title clarifies that we are investigating the impact of COVID-19 related administrative restrictions on spine surgery practice and outcomes in an urban healthcare setting. Importantly, our hospital began their administrative restrictions in March, and as such the collected data represent the results of the first and most severe COVID-19 related surgical rationing.

Abstract:  Comprehensive ,well return & summarises the overall paper very effectively .

Introduction  : Adequate, directed and well organised. No modifications recommended. The lacunae in the literature & the need for the study is clearly written .

Thank you.

Material methods : main concern is the time period of the study as it does not cover the main covid period . The authors have looked into all the aspects including surgical duration , immediate & late complications & outcome . The methodology is good

Thank you. Please see previous comments and modifications related to study period relevance. 

Results :  This section is very well written & the tables are easily understandable. The study also has found no difference with regards to 90 day complications . What is surprising is that there was no difference in the operating time duration . With lot of precautions being taken it is quite natural to presume that the operating time duration would be more .

Thank you. Although we are unable to verify this with data, our hypothesis is that that there was substantial pressure from hospital administration for fast OR turnover as a result of the surgical rationing, and have anecdotally confirmed this with our surgeons. Their perspective was that the administration encouraged high efficiency in the OR so that more essential surgeries could be performed.

Discussion :

Adequate review of the published literature . One of the limitation is the time frame of the study which needs to be mentioned .

Thank you. We have now included an expanded discussion of the relevance of the study time frame in the discussions/limitation section.

Overall the study would have been more of value if it would have covered the overall duration of the epidemic

See previous comment responses for additional context on chosen study period

Reviewer 2 Report

Present a hypothesis after stated purpose.

On lines 98-99, it is stated short-term and long-term follow-up periods in a range form. Please state what the actual follow-up periods ended up being in the results (in a sentence statement).

Line 142 - p-value indicates 0.064, which would mean NOT significant. I believe this is a typo.

Lines 159-160 - states how ER and tele visits increased. However, in discussion (lines 182, 224 and abstract) the opposite is stated. Please correct discussion to match data. 

Line 198 - add reference "[23]" at end of sentence.

Lines 207-209 - this is hypothesis, add to intro.

Loss to follow-up - please state No. per group in results.

Lie 253,254 - Again,  mis-statement about findings. "no difference in complication rates" - This is not true!!

Author Response

 Present a hypothesis after stated purpose.

We have now added a hypothesis statement in the purpose section

On lines 98-99, it is stated short-term and long-term follow-up periods in a range form. Please state what the actual follow-up periods ended up being in the results (in a sentence statement).

We have now specified the follow up periods in a sentence statement in the results section.

Line 142 - p-value indicates 0.064, which would mean NOT significant. I believe this is a typo.

Thank you. We have clarified in the text that this p-value is a non-significant trend, and have removed the indicator for this category in figure 1 to reduce confusion.

Lines 159-160 - states how ER and tele visits increased. However, in discussion (lines 182, 224 and abstract) the opposite is stated. Please correct discussion to match data.

Thank you for bringing this inconsistency to our attention. We have double checked our analyses and corrected the data to reflect the results. Our data demonstrate that although complication rates were lower during the COVID-19 pandemic, this was not a statistically significant reduction. We did observe that when there were complications, patients were more likely to address those complications through ER visits and telemedicine during the COVID-19 pandemic. This has been clarified in the results, discussion, and conclusion sections.

Line 198 - add reference "[23]" at end of sentence.

Thank you, this is now added.

Lines 207-209 - this is hypothesis, add to intro.

We have now consolidated these hypotheses in the introduction

Loss to follow-up - please state No. per group in results.

Thank you. The proportion of patients that were lost to follow up was equivalent between the pre- and during-COVID groups. This is now indicated in the results.

Lie 253,254 - Again, mis-statement about findings. "no difference in complication rates" - This is not true!!

Thank you. We have double checked that this statement is accurate and have clarified our language about complication rates throughout the manuscript.

Round 2

Reviewer 1 Report

Thank you for the clarification & response to the queries raised 

Reviewer 2 Report

The modifications made have greatly improved the manuscript.